# Unequal distributions of crowdsourced weather data in England and Wales

Oscar Brousse [1] ✉, Charles H. Simpson [1], Ate Poorthuis [2] &
Clare Heaviside [1]

Personal weather stations (PWS) can provide useful data on urban climates by densifying the number of weather measurements across major cities. They do so at a lower cost than official weather stations by national meteorological services. Despite the increasing use of PWS data, little attention has yet been paid to the underlying socio-economic and environmental inequalities in PWS coverage. Using social deprivation, demographic, and environmental indicators in England and Wales, we characterize existing inequalities in the current coverage of PWS. We find that there are fewer PWS in more deprived areas which also observe higher proportions of ethnic minorities, lower vegetation coverage, higher building height and building surface fraction, and lower proportions of inhabitants under 65 years old. This implies that data on urban climate may be less reliable or more uncertain in particular areas, which may limit the potential for climate adaptation and empowerment in those communities.

Cities are complex environments with substantial environmental heterogeneities that have a direct impact on their local climates. For instance, cities affect local temperatures and related relative humidity, atmospheric circulation, and precipitation regimes[1]. Particular attention has been given to urban heat because temperature is a reliably measured weather variable, known to have negative impacts on health, buildings and infrastructure, energy, or biodiversity[2–6]. Considering public health alone, hotter urban areas are subject to higher levels of heat-related mortality and morbidity, such as risk of stroke, exhaustion, and cardiovascular diseases[7,8]. Therefore, understanding which urban environments are most associated with extreme heat, and the underlying vulnerabilities of their inhabitants, is necessary to establish sustainable and healthy cities. Networks of official weather stations, such as the national weather station network (MIDAS, Met Office Integrated Data Archive System) managed by the United Kingdom (UK) Met Office, may be too sparse to respond to this specific need. Currently, only 155 MIDAS stations of high accuracy corresponding to World Meteorological Organization (WMO) and Climate Network standards record weather measurements at hourly time steps over the whole of England and Wales, covering an area of 151 139 km²

(Supplementary Fig. 1). To address this data scarcity new sources of weather data are needed.

Over recent decades, the number of weather devices operated by independent individuals who openly share the data collected as a crowdsourcing activity[9] has rapidly increased in European countries, including the UK[10]. We refer to these devices as personal weather stations (PWS) to contrast them with official automatic weather stations installed by meteorological offices. For instance, during the extremely hot summer of 2022, 5011 PWS of the Netatmo brand recorded temperatures, greatly extending the existing coverage of official temperature sensors (see Methods; Supplementary Fig. 1). PWS are therefore increasingly being used in urban climatological studies and are sought to provide useful complementary data sets for urban climate services[11–17]. To deal with higher degrees of uncertainty due to their inaccuracy[18,19] and human factors such as sub-optimal placement, multiple filtering algorithms have been developed to make this data more reliable[20–23]. PWS crowdsourced weather data are quickly becoming suitable and valuable for urban temperature studies and evaluation of models[24]. Pioneering studies have even suggested that they could be used to develop city-wide heat alarm systems[25].

[1]University College London, Institute of Environmental Design and Engineering, London, UK. [2]Katholieke Universiteit Leuven, Department of Earth and Environmental Sciences, Leuven, Belgium. ✉e-mail: o.brousse@ucl.ac.uk

Little attention has however been given to the underlying socio-economic and environmental characteristics of the areas in which PWS are installed. In other words, even though PWS offer an unprecedented opportunity to study urban heat in places that are deprived of official sensors, they may well be unequally distributed among the variety of urban environments that exist in towns and cities. This phenomenon, named the "sensor desert", has already been highlighted for other types of sensors, such as for air pollution[26,27]. Because sensor deserts prevent the acquisition of weather data in a representative way (e.g., covering all communities existing in the country), the generalization of urban heat impact studies is limited. Furthermore, studying the heat exposure across populations with different degrees of vulnerability is hindered. Understanding where PWS are positioned at a national level and the implications for urban climate studies and their related impact studies is therefore important, especially when assessing the suitability of various heat adaptation and mitigation strategies. After all, such networks could help to provide adequate guidance to decision-makers to define national and sub-national guidelines to address the increasing threats caused by extreme heat in inhabited areas as the climate warms.

In this study, we empirically investigate the current coverage of certain PWS that are widely used by consumers and researchers and that are capable of measuring temperature across England and Wales. To do so, we: gather data on the existing coverage of commonly used PWS from the Netatmo company across England and Wales during the record-breaking summer of 2022; collect information on the urban environment via a set of satellite earth observations; and pick out key demographic and socio-economic indicators from the 2011 census in England and Wales (see Methods). We then analyse how each of these indicators is associated to the presence or absence of PWS. Our study provides a comprehensive description of the underlying spatial inequalities in crowdsourced climate data acquisition and accessibility in English and Welsh urban environments.

## Results

We find that more deprived areas, meaning places with the highest Index of Multiple Deprivation score (see subsequent subsection), are generally less covered by PWS than wealthier ones. We also find that, typically, more densely built areas with higher buildings, lower vegetation coverage and albedo (reflective capacity), and areas with higher proportions of ethnic minorities have lower PWS coverage. Although areas with higher proportions of adults over 65 years old, and therefore most vulnerable to heat mortality and morbidity[28–30], are associated with increased PWS coverage, the current distribution of PWS may limit the study of the urban climate and its impact in areas where populations are the most deprived and have a high proportion of ethnic minorities. We provide more details on these results below.

### Levels of deprivation and sensor coverage

Empirical evidence of an unequal distribution of PWS throughout England and Wales is given by the relative number of people per decile of Index of Multiple Deprivation (IMD; Fig. 1) who have at least one PWS in their Lower layer Super-Output Area (LSOA, a kind of census tract). The IMD is a comprehensive index of socioeconomic deprivation, calculated and simplified as deciles at the LSOA level by the UK Office of National Statistics that combines socio-economic and demographic indicators of deprivation (e.g., crime, income, education, etc.)[31]. We use social deprivation here as a proxy for the potential social vulnerability of people to climate hazards[30,32–34]. We find that 24% of people living in the least deprived decile (10th decile) have at least one PWS in their LSOA compared to 3% of those in the most deprived decile (1st decile). For each increase in the IMD decile, we find an increase of ~ 2.3 percentage point in the number of people with at least one PWS in their LSOA. Interestingly, we also find that official automatic weather stations from the MIDAS network are less present, if not absent, from most deprived LSOAs (see Supplementary Fig. 2). At the time of writing, 13.32% of the population of England and Wales is covered by at least one PWS while only 0.5% is from a MIDAS weather station. This

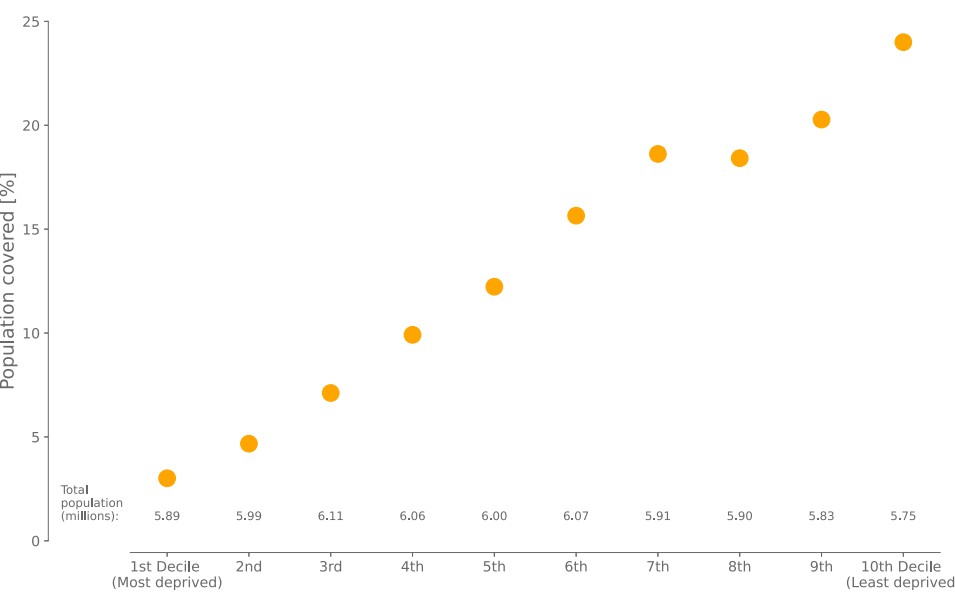

**Personal weather station coverage per deciles of Indice of Multiple Deprivation**

**Fig. 1 | Percentages of people in England and Wales covered by at least one personal weather station (PWS) per Lower layer Super-Output Area (LSOA), per Index of Multiple Deprivation (IMD) decile.** Least deprived populations are better covered by PWS. The total number of people in each IMD decile is given at the bottom of the figure and shows fluctuation due to some variability in LSOA population size (~1500 inhabitants per LSOA; see Methods). The same figure is given in Supplementary Fig. 2 for official automatic weather stations from the Met Office Integrated Data Archive System (MIDAS) network handled by the United Kingdom Met Office and also shows that most deprived populations are less covered by highly accurate weather stations.

highlights the extension of weather sensing networks in a variety of environments offered by PWS and supports the need to describe them.

## Inequalities in the urban-rural continuum

Local Climate Zones (LCZs) are land-use land-cover categories specifically designed for urban climate studies[35], with LCZ 1 to 10 representing built-up classes, and A to G natural classes (see Methods). In England and Wales, urban LSOAs are mostly composed of Compact Mid-Rise (LCZ 2), Open Mid-Rise (LCZ 5), Open Low-Rise (LCZ 6), Large Lowrise (LCZ 8) and Sparsely Built (LCZ 9) environments. Rural LSOAs are composed of Dense Trees (LCZ A), Scattered Trees (LCZ B), Low Plants (LCZ D), Bare rock or pave (LCZ E) and Water (LCZ G) – the latter can be typical of coastal areas. When looking at the density of PWS over the LSOAs' area, we find that LSOAs that are mostly composed of mid-rise buildings have better PWS coverage than others with ~ 0.5 PWS per km² in LCZ 2 and a peak of 0.64 PWS per km² in LCZ 5 (Fig. 2a). Natural areas tend to have a lower proportion of their total area covered by a PWS with ~0.03 PWS per km² on average; no PWS are present in LCZ E. Built-up LCZs, which compose only 11 362 km² of England and Wales (7.5 % of the total 151 139 km²), hence concentrate most of the PWS. The latter confirms that PWS could be regarded as a valuable palliation to the characteristic climate data scarcity in urban areas[10,36].

Yet ultimately, because PWS density may simply be related to population density, it is important to estimate how many individuals are covered by one sensor to define what is the optimal density of weather sensors for capturing the variability of population's exposure to heat. We find the highest coverage per capita in LCZ 9 environments, with 35.9% of its population covered by at least one PWS (Fig. 2b). Overall, 7.9 M inhabitants are covered by a PWS out of the total population of 59.5 M living in England and Wales. Of these, 4.2 M out of the 42.9 M inhabitants that live in mostly built-up LSOAs are covered by PWS in comparison to the 3.7 M out of the remaining 16.6 M inhabitants that live in natural areas. In other words, ~ 22% of the population in natural areas has at least one PWS in their LSOA, but for built-up areas this is only the case for ~ 10% of the population. This suggests that there is an urban-rural inequality in PWS coverage that cannot be solely explained by the larger size of the LSOAs in more rural lands. Indeed, as LSOAs are made of the same approximate amount of inhabitants, the probability of a citizen to buy one PWS should be the same across all environments.

## An urban and demographic sensor desert

We found the existence of an unequal distribution of PWS between urban and natural areas, and between more and less deprived locations in England and Wales. We now further investigate the underlying risk characteristics of each LSOA. In fact, a wealthier urban LSOA may well be at higher risks of heat-exposure than a more deprived rural one because of the urban heat island effect[1,37] and therefore require more PWS coverage to survey the local climate and represent local population exposure to heat. Referring to the risk triangle[38], we therefore need to estimate how likely is the hazard of higher air temperatures in each environment, and how vulnerable the population is to this heat.

Using a set of satellite-based environmental indicators that impact the micro-scale air temperature, or the hazard (see Methods; Fig. 3 and Supplementary Fig. 3), we show that lower proportions of populations are covered by a PWS in environments that have lower vegetation coverage (8.1% in the lowest decile against 24.9% in the highest decile) and higher building heights (8% against 29.3%) and building fractions (6.9% against 30.1%). This could explain the lower shortwave albedo where PWS are absent (9.2% against 26.4%)—asphalted and other dark built surfaces having lower albedos than natural areas on average. Concerning the vulnerability, we find that lower proportions of the total population are covered by a sensor when an LSOA has a lower proportion of people older than 65 years (9.1% against 19.3%) and when ethnic minorities are more present (5.4% against 18%). Overall, this

means that people in England and Wales who live in areas with the highest vegetation coverage and the lowest proportion of ethnic minorities are ~ 3 times more likely to have a PWS measuring their local climate. PWS sensor deserts in England and Wales thereby consist of places with higher buildings and building fractions, deprived of vegetation, with a lower solar radiation reflective capacity, and where younger and more ethnically diverse people live. These demographic and environmental characteristics of sensors deserts also describe lower IMD deciles (Fig. 4, Supplementary Table 1). For example, most deprived LSOAs have higher building heights by ~ 4 m than least deprived LSOAs on average (Supplementary Table 1), building fractions by ~ 10%, and proportion of ethnic minorities by ~ 15%. Also, lower albedo is observed by ~ 0.02 (2% less reflective capacity), EVI by ~ 0.06 (less densely vegetated), and proportion of people aged 65 or above by ~ 10%.

## No deserts are found at similar levels of deprivation

Importantly, we find that within the population living in each IMD decile no clear distinction between the PWS and no-PWS (absence of PWS) can be made out of the environmental and demographic indicators. For example, the highest proportion of people living in the highest IMD decile (10th) live in the 3rd decile of building height (low buildings) independent of whether PWS are present or absent in their LSOA (Fig. 4, Supplementary Table 1). This is further confirmed by calculating the Perkins Skill Scores (PSS) for each covariate in each IMD decile. The Perkins Skill Score[39] is a metric ranging from 0 to 1 that estimates how similar probability density functions are; a score above 0.7 meaning that the probability density functions are generally overlapping. Here we compare the probability density functions between presence and absence of PWS, weighted by the total population in each LSOA, and find that in all IMD deciles, environmental and demographic indicators of the PWS and no-PWS groups always overlap with PSS scores above 0.70—apart from the Enhanced Vegetation Index in the lowest IMD decile (PSS = 0.69; Supplementary Table 1).

In general, this confirms that no clear empirical signals are found that would distinguish places with and without PWS within each IMD decile, and that the links between environmental and demographic characteristics could equally be described by the levels of deprivation, or even explained by them. As higher PSS are found in highest IMD deciles, this suggests that greater dissimilarities between environments covered and deprived of PWS exist in more deprived LSOAs. However, this study does not try to explain the causes of the distribution of PWS across England and Wales but rather characterises environments with the varying PWS coverage. We simply note that PWS coverage varies with area deprivation, and that the environmental and demographic characteristics also vary with area deprivation. Therefore the thermal environments experienced in more deprived areas may not be well studied.

## Discussion

In this study, we use a simple method and easily accessible socio-economic, demographic, and environmental data over England and Wales to highlight an uneven coverage in crowd-sourced weather data derived from individually acquired personal weather stations (PWS). Despite our focus on England and Wales, our analysis proposes a transposable method to other region that offers indicators of socio-economic and demographic deprivation, ours being freely downloadable in England and Wales[40]; the environmental data is globally and freely downloadable[31,41,42]. Our study highlights these inequalities and, thereby, nuances the vision that openly accessible and interconnected personal devices—often called the Internet of Things[43]—could unequivocally mitigate institutional data scarcity in climate studies.

In face of the increasing demand for denser weather data to build liveable and sustainable environments and the parallel

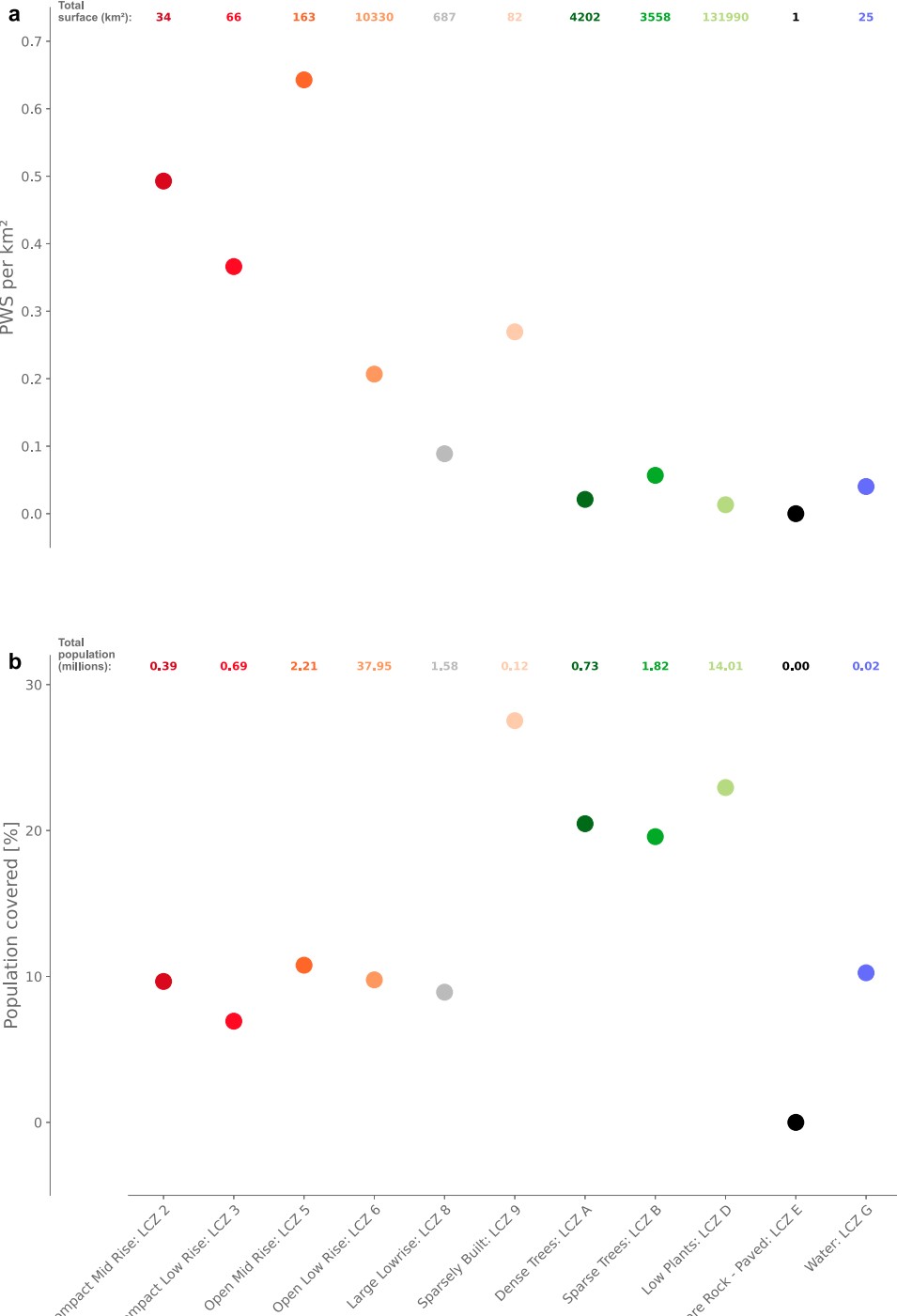

**Fig. 2 | More personal weather stations (PWS) are present in cities but more people are covered by at least one PWS in rural areas.** PWS per square kilometre (km²; **a**) and percentage of total population covered by a PWS (km², **b**) in each Local Climate Zone (LCZ). LCZs are landuse land-cover classes specifically designed for urban climate studies. Total surface covered by each LCZ in England and Wales is given at the top of panel **a** and the respective total population at the top of panel **b**.

expansion in PWS coverage, PWS are seen as an unprecedented opportunity to investigate the impact of urban areas on the local climate and temperatures[36,44] Notwithstanding this opportunity, our study highlights the representativeness and generalizability of data generated by PWS. Local and national policies and action plans can protect residents from extreme heat but require identification of locations and populations that are most at risk[45]. Even if cities were to integrate heat mitigation measures such as changes to building and urban design that would consider local realities in terms of heat hazards and vulnerabilities only certain neighbourhoods and built-up environments would be able to offer the necessary climatic data.

Personal weather station coverage
per deciles of environmental and demographic metrics

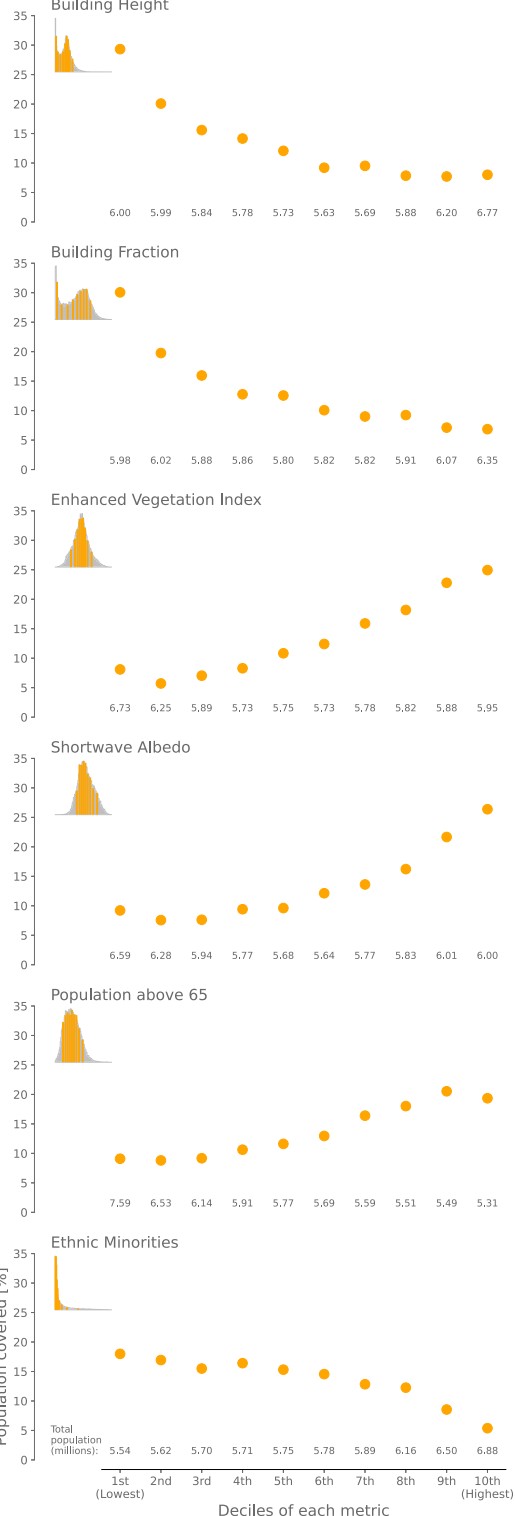

**Fig. 3 | Percentages of population covered by at least one personal weather station (PWS) in each decile of key environmental and demographic metrics for heat-related health risks.** From top to bottom: building height, building fraction, Enhanced Vegetation Index (EVI; proxy for vegetation coverage), short-wave albedo, population above 65 years old and proportion of ethnic minorities. For all metrics, the first decile indicates the group with the lowest values and the tenth the highest values. The total population represented in each decile in given at the bottom of each panel. On the top left corner of the panels, histograms of each metrics are plotted (in grey) to give a sense of the data distribution and where deciles cut-offs are located (vertical orange lines)—full size histograms are provided in the supplements (Supplementary Fig. 3).

Furthermore, the sensor desert may limit quantification of the cooling induced by vegetated evapotranspiration, the latter being shown to have positive impacts on heat mitigation and related heat-mortality[46]. Lastly, ethnic minorities are often more exposed to asphalted built-up areas and related surface urban heat islands[47,48]. This could drive disparities in heat exposure, especially considering that ethnic minorities tend to comprise younger populations and that people aged above 65 tend to be more susceptible to heat-related health hazards[29]. It would therefore be necessary to adequately quantify how vulnerabilities induced by deprivation may increase the risk to different communities, stratified by age groups.

Fortunately, places with higher concentrations of persons over 65 years old tend to have good PWS coverage. But proportions of elderly people are still high (above 10%) in the most deprived areas where sensor deserts are observed. Hence, increasing weather sensors in these locations is paramount to adequately respond to urban climate challenges. Combined with additional vulnerabilities associated with deprivation, including but not limited to obesity, diabetes, workplace conditions, clothing or accessibility to vegetated cool islands[49], these people could be more exposed to heat-related risks, which poses some important ethical issues.

Responses to increased coverage of these areas might include developing parallel weather station networks or incentives to purchase PWS by local populations. To do so, future research should try to understand how different demographic, socio-economic, cultural and environmental covariates may explain the distribution of PWS coverage. Focal population surveys and large-scale machine learning could shed further light on the preliminary findings of our study. For instance, consumer surveys should try to gather information at the individual level to relate how consumers' demographic and socio-economic characteristics are similar or dissimilar to those of the area in which they live (e.g., distance to the average annual income in the LSOA). It would also be necessary to disentangle how deprivation explains PWS coverage because other covariates (e.g., built-up density, population density, or building height) may also have an influence on PWS coverage that we could not separate from deprivation in this study. We nonetheless found that despite some similarities in the distributions of PWS presence across our set of environmental and demographic characteristics—considered central for urban heat-related risks—some heterogeneities still exist and need to be explored. Most importantly, we found that greater differences exist between environments that are covered and not covered by a PWS between the most deprived areas, with smaller differences in the least deprived locations. This could suggest a greater diversity of demographic, socio-economic and environmental characteristics in more deprived areas that need to be investigated to better understand the mechanisms explaining PWS coverage.

We acknowledge that our study suffers from several limitations and encourage further research to address them. First, its conclusions are specific to England and Wales to benefit from a standardized Index of Multiple Deprivation between the two countries—something that is more complex at the supra-national level (e.g., Europe or global) and

An important outcome from our study is the empirical demonstration that denser, poorly vegetated, and deprived built-up areas with higher proportions of ethnic minorities and of people under 65 years old lack crowd-sourced PWS data. This has several implications because denser built-up areas with higher building heights and lower surface albedo are often hotter due to higher radiation absorption and trapping, but are not adequately covered by weather sensing devices.

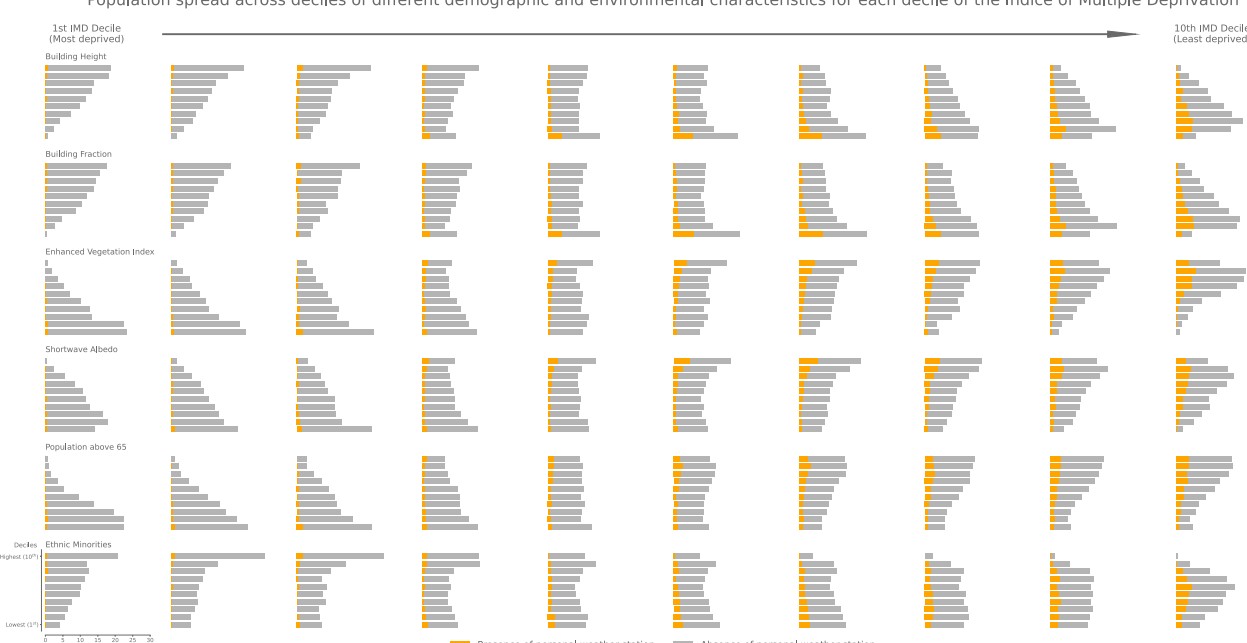

**Fig. 4 | Histograms of the percentages of the total population living in each Index of Multiple Deprivation (IMD) decile across deciles of different environmental and demographic metrics.** The population covered by a personal weather station (PWS) is given in orange while the rest of the population is given in grey (absence of PWS). For all metrics, the first decile indicates the group with the lowest values and the tenth the highest values. Deciles cut-off values and distribution of values per variables can be found in Supplementary Fig. 3. The total population living in each IMD decile is given in Fig. 1.

---

even at national level of the countries of the United Kingdom—and should thus be carefully considered for other locations. Challenges related to the level of aggregation of the socio-economic, demographic and environmental data will have to be addressed to study the factors influencing the distribution of PWS at larger scales. For example, our presence/absence framework would be irrelevant at Nomenclature of Territorial Units for Statistics (NUTS) administrative units as all would be covered by at least one PWS in England and Wales (see Supplementary Fig. 4). Second, because of data restriction policies our study only gathered information from Netatmo PWS and did not investigate other crowd-sourced weather data providers like the Weather Observation Website (WOW)[50] or Weather Underground (Wunderground) which could cover areas deprived of Netatmo PWS[51]. Third, due to data restrictions we did not study the multivariate intersections that might have further implications for heat-related risks, like the proportion of elderly people from ethnic minorities living in mostly asphalted areas. Fourth, as touched above, we did not study how each covariate is correlated and explicative of another—something that would help in guiding efficient campaigns for weather data collection and relevant mitigation and adaptation policies. Fifth, we constrained our analysis to a very small sample of explanatory variables of heat-related risks and did not look at other common variables like the proportion of young children below five or the presence/absence of water bodies[52–55]. Lastly, we assumed that each sensor would be fully operational and did not look at the data completeness and uncertainty, something that can now be achieved via common filtering tools[22,23]. Census data used was from 2011, as the 2021 census data were not yet available. Further work is required to unpick what the effect of these sensor deserts is on measurement uncertainty.

As a final note, we recognise that even if these data scarcities would be dealt with at the national level in countries that already offer a substantial PWS network, global inequalities would still be much greater. For instance, it is commonly accepted that the African continent is underprivileged for the acquisition and the installation of weather stations[56,57] despite the gravity of local climate hazards and inherent challenges that the countries face. Projects like the Trans-African Hydro-Meteorological Observatory (TAHMO)[58] exemplify ambitious and worthwhile programs that try to reduce these global weather data inequalities. However, our study shows that individually acquired personal weather stations do not unequivocally address the limitations associated with these urban sensor deserts and such stations in isolation are thus likely not a sufficient answer to urban climate challenges faced in these countries.

## Methods

### Data acquisition

In this section, we provide details on the choices of data that was acquired for this study (Fig. 5), their origin and what each represents.

### Netatmo personal weather station

We gather all the necessary Netatmo PWS information using the "*GetPublicData*" API via the *patatmo* python module, namely latitude, longitude, and sensor ID. We use a moving window of 0.2° by 0.2° over England and Wales. This step ensures that all Netatmo PWS are captured by the API, since over large domains a reduced number of PWS is accessible via the API. We ran our data collection on the 22nd of October 2022.

After all data has been collected, duplicates are filtered out and only PWS that record temperature are kept. This means that PWS recording only pressure, humidity or precipitation are not included in this study. We collect data on PWS only for those active during the recent summer 2022 (June to August) as this was a record-breaking year in daily maximum temperature in England and Wales with temperatures going as high as 40.3 °C[59]. This also ensures that a maximum of PWS are included in the analysis since their number has kept increasing over the past decade[10].

### Demographic and indices of multiple deprivation

To understand the potential implications of PWS desert coverage, we focus on three major demographic and socio-economic indicators which we justify below: the proportion of people over 65 years old, the

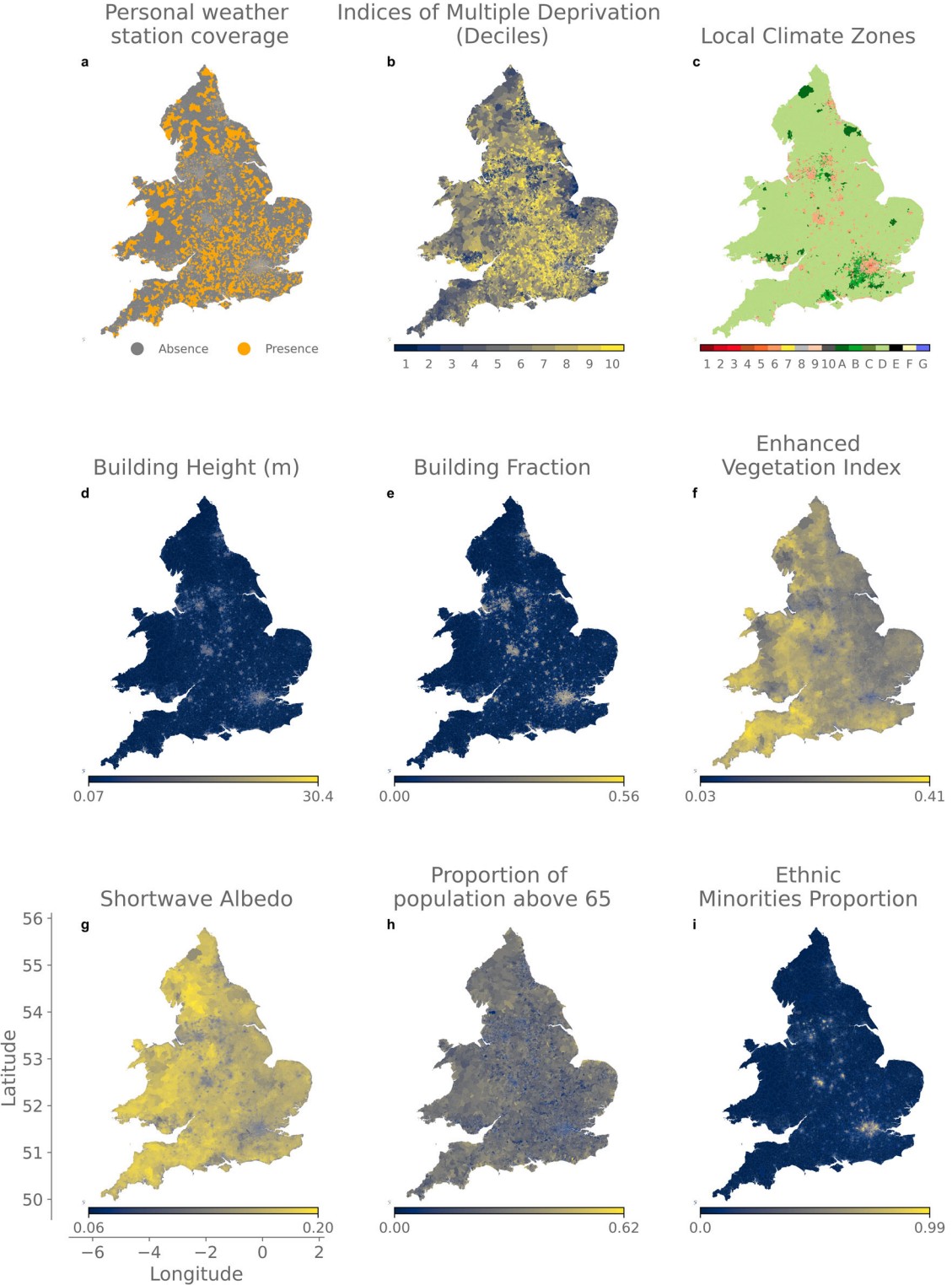

**Fig. 5 | Maps of personal weather stations (PWS) coverage and various factors related to heat exposure and vulnerability in England and Wales.** Presence or absence of PWS (**a**) at the Lower layer Super-Output Areas (LSOA; areas that include ~1500 inhabitants) level and related socio-economic, demographic, and environmental indicators correlated to heat-related risks in England and Wales (**b** to **i**).

proportion of ethnic minorities, and the Index of Multiple Deprivation. In the United Kingdom, people over 65 have been shown to be more vulnerable to the negative impacts of extreme heat especially mortality[60]. Recent studies have demonstrated that certain ethnic minorities are more exposed to higher surface temperatures and are usually further from a vegetated area where they could find cooler temperatures[47]. The Indices of Multiple Deprivation (IMD) developed in England and Wales[31] combine multiple indicators of socioeconomic deprivation, which is thought to be indicative of people's vulnerability to climate hazards[45]—previous studies have found links between low socioeconomic status and risk of death during a heatwave[61]. This is thought to be caused by factors such as quality of housing and general

health. We collect this information at the Lower layer Super-Output Areas (LSOA) for both England and Wales on the official repositories of the Office of National Statistics[40]. LSOAs consist of areas that must include 1500 inhabitants and are thereby dependent on population density for their shapes and sizes. Before data treatment, both Wales and England data were joint using dedicated python packages and common naming was attributed to each variable.

## Environmental earth observations

To determine which type of environment surrounds each PWS we use several surface indicators which are considered important for urban heat variability. First, we collect building surface fraction and building heights from the Global Human Settlement Layer project provided by the European Commission[42] because building presence and height will directly influence the radiation trapping and the production of anthropogenic heat fluxes in urban areas. The two are expected to change the surface energy balance and therefore lead to different urban air temperatures. Second, we derive information on the vegetation coverage by using the Moderate-Resolution Imaging Spectroradiometer (MODIS) MOD09GA v6 Enhanced Vegetation Index (EVI) product at 500 m horizontal resolution[62]. Vegetation density, type and fraction are indeed important contributors to the spatio-temporal variability of heat and is also an indication of the built-up surface fraction and accessibility to vegetated areas. We chose to use the EVI over the Normalized Difference Vegetation Index as it reduces the influence of the atmosphere and the canopy background signal on the measured index while having an increased sensitivity to the leaf area index−hence facilitating the discretisation of different vegetation types in high biomass locations (like forests)[63,64]. Lastly, we obtain information on the short-wave surface albedo from the MODIS MCD43A3 v6 product[41] to explore the capacity of each environment to reflect incoming solar radiation. This is important as certain built-up areas may have higher albedos due to reflective roof coating, for example and thus be cooler than their neighbouring environments. We consider a 5-year median for albedo and EVI (2018-2022), which we consider representative of recent PWS deployment in England and Wales, of the other environmental dataset (LCZ, built-up surface, and height) and of the generic environmental conditions of the summer 2022.

## An urban climate land-use and land-cover

In parallel, we also obtain information on the land-use and land-cover by using the Local Climate Zones (LCZ) European map[65], developed within the World Urban Database and Access Portal Tool (WUDAPT)[66]. The LCZ were mapped at a resolution of 100 m with an overall accuracy above 80% for the whole of Europe making it a valid land use land cover map for studying urban climate related issues in the region; the overall accuracy, the Kappa score and the true positive rates for evaluating how the urban areas' extension were consistent with other products like ESA CCI are also above 90% for the UK[65]. LCZ are land-use and land-cover classes specifically designed for the study of the urban climate and consist of continuous urban and natural environments of several hundreds of meters to few kilometres[35]. They distinguish each urban and natural environments archetypes by their built-up density, building height, building types and uses, and natural coverage. Although the other variables given above are continuous variables and more indicative of each environment's specificities than categorical LCZ, we still include LCZ in our set of variables to show the actual coverage of PWS per LCZ and explore if certain LCZ are deprived of PWS. This is of importance for urban climatological and meteorological studies as LCZ are standardized classes that permit the communication of urban climate studies outcomes globally.

## Data post-treatment

Once we gathered all the necessary data, we aggregated all the necessary information at the LSOA level. Therefore, for each LSOA we:

(i) used the *rasterio zonal stats* python package to estimate the average values of building surface fraction, building heights, surface albedo and EVI in the LSOA; (ii) used the same package to get the modal LCZ in the LSOA (see Supplementary Discussion, Supplementary Fig. 5 and Supplementary Fig. 6 for more information on the potential loss of information induced by the use of modal LCZ); (iii) calculated the proportion of people aged above 65 years old against the total LSOA population; and (iv) counted the proportion of ethnic minorities per LSOA by considering all classes that are not "*White: English/Welsh/Scottish/Northern Irish/British*" as minorities and counted them against the total LSOA population. Indices of multiple deprivation are left in their decile format.

## Inclusion and ethics statement

We recognise that ethnic groups are not biologically founded and are sociopolitical constructs to which not all individuals identify. The groups used in this study are those used during the 2011 Census from the English and Welsh Office for National Statistics. Since 1991, at the time of new census (every 10 years), ethnic groups are defined in consultation with: the users of the census data, such as local institutions and health services; people who complete the census and their representative interest groups; and the National Records of Scotland as well as the Northern Ireland Statistics and Research Agency who are responsible for running the censuses in their respective home nations. The ethnic groups used in this study were not designed by the authors. As the vast majority of individuals living in England and Wales identified as "*White: English/Welsh/Scottish/Northern Irish/British*", we qualified all the other ethnic groups as minorities.

## Study's rationale and framework

In our study, we decided to adopt a simple but robust approach to depict the unequal distribution of PWS amongst the variety of inhabited environments that compose England and Wales. Most importantly, we chose not to perform any explanatory variable analysis because we are not seeking to causally explain the drivers of coverage in PWS, but rather to describe the current state and the implications for sensing of the environment. Moreover, explanation of the drivers of PWS acquisition would be much better addressed by individual data rather than area-level data, for example through surveys or interviews. Machine-learning based studies attempting to explain the distribution of PWS between areas may be obscured by the spatial correlation between PWS coverage and urbanization, which would therefore require extensive studies and analyses that cannot solely be addressed in this paper.

We focused our analysis to the presence or absence of PWS in an LSOA as very few LSOAs have more than one PWS, which prevents any calculation of the relationship between PWS density and the other covariates. Of the 34632 LSOAs composing England and Wales, 3863 are covered by one PWS, and 539 contain more than one; 434 having only two. We therefore assume that the population of an LSOA is covered by a PWS if there is at least one in the LSOA. Because each LSOA comprises a different number of inhabitants, we weight by LSOA population. This means that the numbers presented in this study simply show how many people are covered or not by a PWS.

## Reporting summary

Further information on research design is available in the Nature Portfolio Reporting Summary linked to this article.

# Data availability

All data used in this study have been uploaded on a public GitHub repository. We collected the data as follows: The European continental LCZ map was directly downloaded from the World Urban Database and Access Portal Tool website. MODIS Terra Daily Enhanced Vegetation Index (EVI) was obtained from Google's Earth Engine.

MCD43A3.061 shortwave MODIS Albedo Daily 500 m was also obtained from Google's Earth Engine here. Global Human Settlement Layers built-up surfaces (GHS-BUILT-S) and built-up heights (GHS-BUILT-H) was obtained from the European Commission website. The ethnic groups were obtained from the Office for National Statistics 2011 census data sheet QS201EW. The population (age and total population per Lower layer Super-Output Area) was obtained from the Office for National Statistics 2011 census datasheet KS102EW. The 2019 Index of Multiple Deprivation (IMD) for England and Wales was obtained from the Consumer Data Research Centre (CDRC) datasets. The Met Office Integrated Data Archive System (MIDAS) metadata was obtained from the Centre for Environmental Data Analysis (CEDA) Archive. We used the secured FTP protocol to download it using WinSCP – this requires an accredited account from CEDA; more info is available here. The Netatmo metadata was obtained using the "*patatmo*" Python API which requires the users to be Netatmo App developers (see https://dev.netatmo.com/). All necessary information and links for installation are available at: https://nobodyinperson.gitlab.io/python3-patatmo/. MIDAS and *Netatmo* stations metadata can be obtained on the dedicated websites and through their relative APIs. The raw *Netatmo* and MIDAS metadata are protected and cannot be shared. Publicly sharable data has been released under the Zenodo repository with https://doi.org/10.5281/zenodo.10950425; the latter is linked to the public GitHub repository. Metadata files used in this study by the authors can be obtained upon reasonable request to Dr. Oscar Brousse (o.brousse@ucl.ac.uk).

## Code availability

All the codes used in this study have been uploaded on the same public GitHub repository (https://github.com/oscarbrousse/NatComms_PWS_2024) than the data and are released under the Zenodo repository with https://doi.org/10.5281/zen-odo.10950425. Codes for obtaining *Netatmo* metada are provided there.

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

## Acknowledgements

We would like to thank the providers of the openly accessible data sets used in this study, namely: the WUDAPT project for the LCZ maps, Google's Earth Engine for the MODIS data, courtesy of the NASA and USGS, the European Commission for the GHSL data, and the Office of National Statistics for the census and the IMD data. Additionally, the authors would like to thank Prof. Nicole van Lipzig who organised a seminar that led to this collaboration and Dr. Daniel Fenner and Dr. Fred Meier for their support in gathering and filtering of Netatmo crowd-sourced data. C.H. is supported by a NERC fellowship (NE/R01440X/1)

and acknowledges funding for the HEROIC project (216035/Z/19/Z) from the Wellcome Trust, which funds O.B. and C.H.S.

## Author contributions

O.B. and C.H.S. conceived this study, O.B. conducted the Netatmo data gathering and curation, O.B. and C.H.S. collected the socio-economic, environmental and demographic data, O.B., C.H.S., and A.P. conducted the data analysis, O.B. led this study. O.B., C.H.S., A.P., and C.H. equally contributed to the review of the study design and to the writing of the manuscript.

## Competing interests

The authors declare no competing interests.
