## [Peer Review File · Nature Communications]

REVIEWER COMMENTS

Reviewer #1 (Remarks to the Author):

I believe this paper is of general interest: the authors have convinced me that crowdsourced weather data have the potential to improve climate adaptation and that it is therefore important to understand the biases in personal weather stations' distribution. The methodology, despite being extremely simple, is sound and adapted to the authors' purpose. I only have a couple of comments on the manuscript.

First, are there biases in the distribution of official weather stations? And if so, how does the distribution of official weather stations compare with the distribution of personal weather stations?

Second, I find some parts of paragraphs 102–111 confusing. Please clarify to which figure or table the results refer. I also do not understand the "true" l108. In addition, doesn't sentence l106-108 ("Hence, regardless...") contradict sentence l102-102 ("Demographic and environmental characteristics are also a reflection of the characteristics that define each IMD decile")? Please state more clearly what the overall message of this paragraph is, as well as the meaning of each table and figure.

"The environmental data being globally and freely downloadable" (l126–127): Please check your references; their formatting doesn't seem right, and some of the references provided are for England only, while your text mentions globally available data.

I appreciate that the authors fully describe the limitations of their study in Discussion (l164).

Reviewer #2 (Remarks to the Author):

Review of NCOMMS-23-12133

Unequal distributions of crowdsourced weather data in England and Wales

By Brousse et al.

Summary: This paper builds on the use of crowdsourced weather observations by private weather stations to study meteorology in areas with sparse official observations, i.e., mainly in cities. The paper studies the availability of weather observations by NetAtmo weather stations in cities and neighborhoods in England and Wales and stratifies these along socio-economic factors at the neighborhood level. It appears that relatively fewer PWS are present in neighborhoods with relatively low incomes, a low green fraction, relatively high buildings and built-up fractions, and a relatively high presence of 65+ people. The claim is that we know less about heat stress in these areas, which are neighborhoods where the urgency is likely relatively high. In my opinion, the research design is well thought out, given the limited amount of information that is known in terms of meta-data from the weather stations. The paper is well structured and well written, with good-quality figures. Nevertheless, I have some remarks that could be taken into consideration.

Recommendation: Minor revisions needed

Remarks:

1. Despite I find the paper's analysis is sound, the result is not so surprising in the sense that a priori it could be predicted that neighbourhoods with low incomes have more urgent problems to take care of than buying weather stations and thinking about heat stress, climate change and climate adaptation.

2. I was wondering whether the gap in private weather stations in the "low income"-areas is not compensated by dedicated/professional weather stations by health agencies or the municipality, since they know these areas are most vulnerable. Could this be studied too to ensure the drawn conclusions in this manuscript are robust?

3. The study is limited to the area of Wales and England. It is not clear why not whole to U.K. was chosen? It seems now a small subset of counties in the home country of the researchers. How this result is generalizable to a wider area remains unclear now. I fully understand it is hard to obtain all socio-economic data across the globe, but at least for Europe this information should be available through Eurostat.

4. An additional stratification that could advance the study, is to study whether the same pattern is seen when a) extra instruments are taken into account (i.e. Netatmo the wind and rain sensor are extra costs, so it could be interesting to know which income threshold or so is needed to buy a wind sensor or rain sensor above the thermometer set), b) instruments from more expensive brands like Davis Vantage Pro (1200 euro instead of 150 euro for a NetAtmo thermometer set).

5. The study does implicitly imply that the owner of the PWS is also the one who bought the PWS, but PWSes are very often offered as a gift for Christmas, anniversaries and so on. So whether a PWS is present in a garden may depend more about the wealth of the ones who offer the PWS than the ones that operate the PWS in the end. How was dealt with that, and how does this affect your conclusions?

Reviewer #3 (Remarks to the Author):

General comments

The main objective of this study is to uncover socio-economic and environmental inequalities in the distribution of crowdsourced personal weather stations. The key findings include a lower presence of crowdsourced personal weather stations (PWS) in deprived environments which can be associated with high build-up, low vegetation cover, high proportions of ethnic minorities and mostly younger people. The study uses straightforward yet methodologically sound empirical techniques to obtain these rather logical but important results.

Major comments

o How do the authors account for their inability to disentangle the impact of environmental and socio-economic factors (such as built-up density, building height, age, and ethnic minority representation) on PWS coverage independently from the influence of deprivation? What strategies might be employed to disentangle the role of deprivation in explaining PWS coverage?

o Why did the authors decide to include the Index of Multiple Deprivation (IMD), Local Climate Zones (LCZs) and a variety socio-economic and environmental indicators, even though they exhibit inherent correlations?

o It is know that coarse vegetation indexes are not ideal to study temperature-environment relationships (Bartesaghi-Koc et al., 2020; Rakoto et al., 2021), since different functional green types may have contradicting cooling effects. In this context, it's important to consider whether the EVI takes into account various functional green types. Could you please clarify how the EVI incorporates information about vegetation density, type, and fraction?

o There appears to be missing information regarding the LCZ map used in the study. Was the study area extracted directly from a broader European LCZ map? Can a Europe-wide LCZ map provide sufficient precision for examining the specific LCZs within England and Wales? Additionally, it would be valuable to know more about the spatial resolution, accuracy, and other relevant characteristics of the LCZ map employed in the research.

o Given that PWSs have demonstrated the capability to capture fine-scale temperature variations, encompassing both inter- and intra-LCZ variability (Fenner et al., 2017; Verdonck et al., 2017), it might be considered a limitation to classify PWSs solely into a major LCZ category based on the predominant LCZ within the corresponding Lower Layer Super Output Area (LSOA). Such an approach may result in the loss of valuable information. It would be beneficial to include details regarding the number of distinct LCZ types within a single LSOA and their respective proportions, providing a more nuanced perspective on the urban microclimate in the study area.

o Building upon the preceding comment, as the entire dataset is aggregated at the LSOA level, it is reasonable to assume that readers would find it valuable to have insight into the characteristics and key metrics of these LSOAs. Providing an overview of the LSOA layout and sharing general metrics for these areas can enhance the reader's understanding of the study context.

o The organization of the Results and Methods sections can be improved for better readability. To achieve this, incorporating subheadings to break down the content into smaller, more manageable sections would be beneficial. Additionally, aligning the flow and structure of both sections could further enhance the overall readability of the manuscript.

Minor comments

o Line 12-13: This sentence should be supported with appropriate citations: "Particular attention has been given to urban heat because temperature is a reliably measured weather variable, known to have negative impacts on health, buildings, energy, or biodiversity."

o Figure 2: From the caption of Figure 2 it is unclear whether the total surface and total population of the different LCZ classes in figure 2a and figure 2b relate to the original LCZs derived from WUDAPT or whether they present the LSOAs with modal LCZ.

o Figure 2: Reference to panel b is missing in the figure caption: "Personal weather stations (PWS) per square kilometre (km²; a) and percentage of total population covered by a PWS (km²; b) in each Local Climate Zone (LCZ)."

o Line 105-106: “Also, higher albedo are observed by ~ 0.02 (2 % more reflective capacity), EVI by ~ 0.06 (denser vegetation), and proportion of people aged 65 or above by ~ 10 %.”

This sentence gives the impression that the most deprived LSOAs are associated with higher albedo, denser vegetation and older people. That seems quite counterintuitive and the opposite is shown in Figure 4:

o Line 112: It is not indicated where the results of the PSS are shown. A reference to Supplementary Table 1 could be included.

o Line 127: Period punctuation between sentences is missing.

o Line 132: Period punctuation between sentences is missing.

o Line 140-142: “Furthermore, the sensor desert may limit quantification of the cooling induced by vegetated evapotranspiration and related benefits to heat mitigation (lungman, et al, 2023).”

The statement is true but the cited reference does not consider PWS in quantifying the cooling potential of vegetation.

o Line 142-145. This sentence is quite long. Consider breaking it down into smaller parts to improve the readability: “Moreover, as people aged above 65 tend to be more susceptible to heat-related health hazards and because ethnic minorities are often more exposed to asphalted built-up areas and their related surface urban heat islands – which could drive disparities in heat exposure – it would be necessary to adequately quantify how vulnerabilities induced by deprivation may increase the vulnerability of different communities, stratified by age groups.”

o Line 156-157: The closing parenthesis seems to be missing: “For instance, it would be necessary to disentangle how deprivation explains PWS coverage because other covariates (e.g., built-up density, population density, or building height may also have an influence on PWS coverage that we could not separate from deprivation in this study.”

o Line 193: This sentence appears to be incomplete: “We use a moving window of 0.2° by 0.2° over England and.”

o Line 261: This reference appears to be incomplete: “Oke, T. R., Mills, G., Christen, A. & Voogt, J. A. Urban climates (Cambridge University Press, 2017)”

o Line 318: This reference appears to be incorrect/incomplete: “for National Statistics, O. Office for national statistics: 2011 census geography products for england and wales.”.

o Line 319-320: This reference appears to be incorrect/incomplete: “Schaaf, C. & Wang, Z. Mcd43a3 modis/terra+ aqua brdf/albedo daily l3 global-500m v006. nasa eosdis land processes daac (2015).”

References

Bartesaghi-Koc, C., Osmond, P., & Peters, A. (2020). Quantifying the seasonal cooling capacity of ‘green infrastructure types’ (GITs): An approach to assess and mitigate surface urban heat island in Sydney, Australia. *Landscape and Urban Planning*, 203(February), 103893. <https://doi.org/10.1016/j.landurbplan.2020.103893>

Fenner, D., Meier, F., Bechtel, B., Otto, M., & Scherer, D. (2017). Intra and inter ‘local climate zone’ variability of air temperature as observed by crowdsourced citizen weather stations in Berlin, Germany. *Meteorologische Zeitschrift*, 26(5), 525–547. <https://doi.org/10.1127/metz/2017/0861>

Rakoto, P. Y., Deilami, K., Hurley, J., Amati, M., & Sun, Q. (Chayn). (2021). Revisiting the cooling effects of urban greening: Planning implications of vegetation types and spatial configuration. *Urban Forestry and Urban Greening*, 64(January), 127266. <https://doi.org/10.1016/j.ufug.2021.127266>

Verdonck, M. L., Okujeni, A., van der Linden, S., Demuzere, M., De Wulf, R., & Van Coillie, F. (2017). Influence of neighbourhood information on ‘Local Climate Zone’ mapping in heterogeneous cities. *International Journal of Applied Earth Observation and Geoinformation*, 62(September 2016), 102–113. <https://doi.org/10.1016/j.jag.2017.05.017>

Response to reviewers for NCOMMS-23-12133: “Unequal distributions of crowdsourced weather data in England and Wales”

Reviewer 1

I believe this paper is of general interest: the authors have convinced me that crowdsourced weather data have the potential to improve climate adaptation and that it is therefore important to understand the biases in personal weather stations' distribution. The methodology, despite being extremely simple, is sound and adapted to the authors' purpose. I only have a couple of comments on the manuscript.

We thank the reviewer for their valuable comments. Our detailed answers are provided below in black and cited text is provided between quotation marks. We have uploaded the codes and the public data used in our study in a dedicated GitHub repository: https://github.com/oscarbrousse/NatComms_PWS_2024

First, are there biases in the distribution of official weather stations? And if so, how does the distribution of official weather stations compare with the distribution of personal weather stations?

We have added a figure in the supplement showing the distribution of active official weather stations handled by the UK MetOffice that respond to WMO standards or are part of the Climate Network and that record weather measurement at hourly intervals. We have changed the text in the introduction of the manuscript which now reads (lines 18 to 28):

“Currently, only 155 MIDAS stations of high accuracy corresponding to WMO and Climate Network standards record weather measurements at hourly time steps over the whole of England and Wales, covering an area of 151 139 km² (Supp. Figure 1). To address this data scarcity new sources of weather data are needed.

Over recent decades, the number of weather devices operated by independent individuals who openly share the data collected as a crowdsourcing activity⁹ has rapidly increased in European countries, including the United Kingdom¹⁰. We refer to these devices as personal weather stations (PWS) to contrast them with official automatic weather stations installed by meteorological offices. For instance, during the extremely hot summer of 2022, 5011 PWS of the Netatmo brand recorded temperatures, greatly extending the existing coverage of official temperature sensors (see Methods; Supp. Figure 1). PWS are therefore increasingly being used in urban climatological studies and are sought to provide useful complementary data sets for urban climate services^{11–17}.”

Presence or absence of: Met Office MIDAS automatic weather station (AWS),
Netatmo personal weather stations (PWS), or both

Supp. Figure 1: LSOAs where official weather stations from the UK MetOffice MIDAS network (AWS) are active are in yellow, where *Netatmo* personal weather stations (PWS) are present are in orange, where both are active are in red (Both) and where none is present are shown in gray (Absence).

We also added lines 67 to 70:

“Interestingly, we also find that official automatic weather stations from the MIDAS network are less present, if not absent, from most deprived LSOAs (see Supp. Figure 2). At the time of writing, 13.32 % of the population of England and Wales is covered by at least one PWS while only 0.5 % is from a MIDAS weather station. This highlights the extension of weather sensing networks in a variety of environments offered by PWS and supports the need to describe them.”

Second, I find some parts of paragraphs 102–111 confusing. Please clarify to which figure or table the results refer. I also do not understand the "true" l108. In addition, doesn't sentence l106-108 ("Hence, regardless...") contradict sentence l102-102 ("Demographic and environmental characteristics are also a reflection of the characteristics that define each IMD decile")? Please state more clearly what the overall message of this paragraph is, as well as the meaning of each table and figure.

We agree with the reviewer that this paragraph was confusing. What we wanted to emphasise in this section is that the overall pattern in environmental and demographic characteristics of places with higher PWS coverage generally matches the environmental and demographic characteristics of places with less deprivation (e.g., less deprived areas tend to have lower

building heights), and that when stratified by IMD decile these patterns are attenuated. We have rephrased the paragraph which now reads (lines 110 to 133):

“These demographic and environmental characteristics of sensors deserts also describe lower IMD deciles (Figure 4, Supp. Table 1). For example, most deprived LSOAs have higher building heights by ~4 m than least deprived LSOAs on average (Supp. Table 1), building fractions by ~10 %, and proportion of ethnic minorities by ~15 %. Also, lower albedo is observed by ~0.02 (2 % less reflective capacity), EVI by ~0.06 (less densely vegetated), and proportion of people aged 65 or above by ~10 %.

No deserts are found at similar levels of deprivation

Importantly, we find that within the population living in each IMD decile no clear distinction between the PWS and no-PWS can be made out of the environmental and demographic indicators. For example, the highest proportion of people living in the highest IMD decile (10th) live in the 3rd decile of building height (low buildings) independent of whether PWS are present or absent in their LSOA (Figure 4, Supp. Table 1). This is further confirmed by calculating the Perkins Skill Scores (PSS) for each covariate in each IMD decile. The Perkins Skill Score³⁸ is a metric ranging from 0 to 1 that estimates how similar probability density functions are; a score above 0.7 meaning that the probability density functions are generally overlapping. Here we compare the probability density functions between presence and absence of PWS, weighted by the total population in each LSOA, and find that in all IMD deciles, environmental and demographic indicators of the PWS and no-PWS groups always overlap with PSS scores above 0.70 – apart from the Enhanced Vegetation Index in the lowest IMD decile (PSS = 0.69; Supp. Table 1).

In general, this confirms that no clear empirical signals are found that would distinguish places with and without PWS within each IMD decile, and that the links between environmental and demographic characteristics could equally be described by the levels of deprivation, or even explained by them. As higher PSS are found in highest IMD deciles, this suggests that greater dissimilarities between environments covered and deprived of PWS exist in more deprived LSOAs. However, this study does not try to explain the causes of the distribution of PWS across England and Wales but rather characterises environments with the varying PWS coverage. We simply note that PWS coverage varies with area deprivation, and that the environmental and demographic characteristics also vary with area deprivation. Therefore the thermal environments experienced in more deprived areas may not be well studied.”

“The environmental data being globally and freely downloadable” (l126–127): Please check your references; their formatting doesn't seem right, and some of the references provided are for England only, while your text mentions globally available data.

We thank the reviewer for noting the formatting of the references. We have dealt with this. For the environmental data we maintain that it is freely and globally accessible; even the data for England can be accessed from anywhere outside of England as long as proper justification is provided to the Office For National Statistics. We have written lines 137 to 139:

“Despite our focus on England and Wales, our analysis proposes a transposable method to any country that offers indicators of socio-economic and demographic deprivation, ours being freely downloadable in England and Wales³⁹; the environmental data is globally and freely downloadable^{31,40,41}. Our study highlights these inequalities and, thereby, nuances the vision that openly accessible and interconnected personal devices – often called the Internet of Things⁴² – could unequivocally mitigate institutional data scarcity in climate studies.”

I appreciate that the authors fully describe the limitations of their study in Discussion (l164).

We greatly appreciate that the reviewer is content with the discussion that we provided in the first manuscript. We have added some discussion to answer concerns presented by the second and third reviewers. Discussion now reads (lines 134 to 208):

“Discussion

In this study, we use a simple method and easily accessible socio-economic, demographic, and environmental data over England and Wales to highlight an uneven coverage in crowd-sourced weather data derived from individually acquired personal weather stations (PWS). Despite our focus on England and Wales, our analysis proposes a transposable method to any country that offers indicators of socio-economic and demographic deprivation, ours being freely downloadable in England and Wales³⁹; the environmental data is globally and freely downloadable^{31,40,41}. Our study highlights these inequalities and, thereby, nuances the vision that openly accessible and interconnected personal devices – often called the Internet of Things⁴² – could unequivocally mitigate institutional data scarcity in climate studies.

Crowdsourcing of weather data may not cope with data scarcity in deprived areas

In face of the increasing demand for denser weather data to build liveable and sustainable environments and the parallel expansion in PWS coverage, PWS are seen as an unprecedented opportunity to investigate the impact of urban areas on the local climate and temperatures^{35,43}. Notwithstanding this opportunity, our study highlights the representativeness and generalizability of data generated by PWS. Local and national policies and action plans can protect residents from extreme heat but require identification of locations and populations that are most at risk⁴⁴. Even if cities were to integrate heat mitigation measures such as changes to building and urban design that would consider local realities in terms of heat hazards and vulnerabilities only certain neighbourhoods and built-up environments would be able to offer the necessary climatic data.

An important outcome from our study is the empirical demonstration that denser, poorly vegetated, and deprived built-up areas with higher proportions of ethnic minorities and of people under 65 years old lack crowd-sourced PWS data. This has several implications because denser built-up areas with higher building heights and lower surface albedo are often hotter due to higher radiation absorption

and trapping, but are not adequately covered by weather sensing devices. Furthermore, the sensor desert may limit quantification of the cooling induced by vegetated evapotranspiration, the latter being shown to have positive impacts on heat mitigation and related heat-mortality⁴⁵. Lastly, ethnic minorities are often more exposed to asphalted built-up areas and related surface urban heat islands^{46,47}. This could drive disparities in heat exposure, especially considering that ethnic minorities tend to comprise younger populations and that people aged above 65 tend to be more susceptible to heat-related health hazards²⁹. It would therefore be necessary to adequately quantify how vulnerabilities induced by deprivation may increase the risk to different communities, stratified by age groups.

Fortunately, places with higher concentrations of persons over 65 years old tend to have good PWS coverage. But proportions of elderly people are still high (above 10 %) in the most deprived areas where sensor deserts are observed. Hence, increasing weather sensors in these locations is paramount to adequately respond to urban climate challenges. Combined with additional vulnerabilities associated with deprivation, including but not limited to obesity, diabetes, workplace conditions, clothing or accessibility to vegetated cool islands⁴⁸, these people could be more exposed to heat-related risks, which poses some important ethical issues.

A need for causal explanations of personal weather sensors' coverage

Responses to increased coverage of these areas might include developing parallel weather station networks or incentives to purchase PWS by local populations. To do so, future research should try to understand how different demographic, socio-economic, cultural and environmental covariates may explain the distribution of PWS coverage. Focal population surveys and large-scale machine learning could shed further light on the preliminary findings of our study. For instance, consumer surveys should try to gather information at the individual level to relate how consumers' demographic and socio-economic characteristics are similar or dissimilar to those of the area in which they live (e.g., distance to the average annual income in the LSOA). It would also be necessary to disentangle how deprivation explains PWS coverage because other covariates (e.g., built-up density, population density, or building height) may also have an influence on PWS coverage that we could not separate from deprivation in this study. We nonetheless found that despite some similarities in the distributions of PWS presence across our set of environmental and demographic characteristics – considered central for urban heat-related risks – some heterogeneities still exist and need to be explored. Most importantly, we found that greater differences exist between environments that are covered and not covered by a PWS between the most deprived areas, with smaller differences in the least deprived locations. This could suggest a greater diversity of demographic, socio-economic and environmental characteristics in more deprived areas that need to be investigated to better understand the mechanisms explaining PWS coverage.

Study's limitations and future recommendations

We acknowledge that our study suffers from several limitations and encourage further research to address them. First, its conclusions are specific to England and Wales to benefit from a standardized Index of Multiple Deprivation between the two countries – something that is more complex at the supra-national level (e.g., Europe or global) and even at national level of the countries of the United Kingdom – and should thus be carefully considered for other locations. Challenges related to the level

of aggregation of the socio-economic, demographic and environmental data will have to be addressed to study the factors influencing the distribution of PWS at larger scales. For example, our presence/absence framework would be irrelevant at Nomenclature of Territorial Units for Statistics (NUTS) administrative units as all would be covered by at least one PWS in England and Wales (see Supp. Figure 4). Second, because of data restriction policies our study only gathered information from Netatmo PWS and did not investigate other crowd-sourced weather data providers like the Weather Observation Website (WOW)⁴⁹ or Weather Underground (Wunderground) which could cover areas deprived of Netatmo PWS⁵⁰. Third, due to data restrictions we did not study the multivariate intersections that might have further implications for heat-related risks, like the proportion of elderly people from ethnic minorities living in mostly asphalted areas. Fourth, as touched above, we did not study how each covariate is correlated and explicative of another – something that would help in guiding efficient campaigns for weather data collection and relevant mitigation and adaptation policies. Fifth, we constrained our analysis to a very small sample of explanatory variables of heat-related risks and did not look at other common variables like the proportion of young children below five or the presence/absence of water bodies^{51–54}. Lastly, we assumed that each sensor would be fully operational and did not look at the data completeness and uncertainty, something that can now be achieved via common filtering tools^{22,23}. Census data used was from 2011, as the 2021 census data were not yet available. Further work is required to unpick what the effect of these sensor deserts is on measurement uncertainty.

As a final note, we recognise that even if these data scarcities would be dealt with at the national level in countries that already offer a substantial PWS network, global inequalities would still be much greater. For instance, it is commonly accepted that the African continent is underprivileged for the acquisition and the installation of weather stations^{55,56} despite the gravity of local climate hazards and inherent challenges that the countries face. Projects like the Trans-African Hydro-Meteorological Observatory (TAHMO)⁵⁷ exemplify ambitious and worthwhile programs that try to reduce these global weather data inequalities. However, our study shows that individually acquired personal weather stations do not unequivocally address the limitations associated with these urban sensor deserts and such stations in isolation are thus likely not a sufficient answer to urban climate challenges faced in these countries.”

Reviewer 2

This paper builds on the use of crowdsourced weather observations by private weather stations to study meteorology in areas with sparse official observations, i.e., mainly in cities. The paper studies the availability of weather observations by NetAtmo weather stations in cities and neighborhoods in England and Wales and stratifies these along socio-economic factors at the neighborhood level. It appears that relatively fewer PWS are present in neighborhoods with relatively low incomes, a low green fraction, relatively high buildings and built-up fractions, and a relatively high presence of 65+ people. The claim is that we know less about heat stress in these areas, which are neighborhoods where the urgency is likely relatively high. In my opinion, the research design is well thought out, given the limited amount of information that is known in terms of meta-data from the weather stations. The paper is well structured and well written, with good-quality figures. Nevertheless, I have some remarks that could be taken into consideration.

We thank the reviewer for their valuable comments. Our detailed answers are provided below in black and cited text is provided between quotation marks. We have uploaded the codes and the public data used in our study in a dedicated GitHub repository: https://github.com/oscarbrousse/NatComms_PWS_2024

1. Despite I find the paper's analysis is sound, the result is not so surprising in the sense that a priori it could be predicted that neighbourhoods with low incomes have more urgent problems to take care of than buying weather stations and thinking about heat stress, climate change and climate adaptation.

We agree with the reviewer and this is why studying the exact nature of this unequal distribution is important. Weather sensor coverage could help designing adaptive and mitigative strategies to cope with data scarcity in order to build more resilient cities in face of local climate changes and extreme heat. We address this point with this introductory sentence (lines 40 to 42):

“After all, such networks could help to provide adequate guidance to decision-makers to define national and sub-national guidelines to address the increasing threats caused by extreme heat in inhabited areas as the climate warms.”

2. I was wondering whether the gap in private weather stations in the “low income”-areas is not compensated by dedicated/professional weather stations by health agencies or the municipality, since they know these areas are most vulnerable. Could this be studied too to ensure the drawn conclusions in this manuscript are robust?

We thank the reviewer for suggesting to study this. Briefly, health agencies and municipalities in the UK generally rely on MetOffice weather stations and do not have their own. But, as answered to reviewer 1, the official coverage of automatic weather sensors is really limited across England when it comes to studying and surveying the local exposure of populations to extreme heat. Across the whole UK only 155 automatic weather stations are installed and monitored by the MetOffice. This is considered a large weather sensor for studying the long-term evolution of the national climate and offers unprecedented levels of accuracy. Nonetheless, the spatial coverage is limited and does not cover major residential areas thus preventing the study of socio-economic disparities in terms of exposure to different

temperatures (average, minimum, maximum or extremes...). We therefore look at the opportunities that new publicly accessible data offer to cope with this research question.

As explained to reviewer 1, we have changed the text in the introduction of the manuscript to highlight the scarcity of official weather stations (lines 18 to 28):

“Currently, only 155 MIDAS stations of high accuracy corresponding to WMO and Climate Network standards record weather measurements at hourly time steps over the whole of England and Wales, covering an area of 151 139 km² (Supp. Figure 1). To address this data scarcity new sources of weather data are needed.

Over recent decades, the number of weather devices operated by independent individuals who openly share the data collected as a crowdsourcing activity⁹ has rapidly increased in European countries, including the United Kingdom¹⁰. We refer to these devices as personal weather stations (PWS) to contrast them with official automatic weather stations installed by meteorological offices. For instance, during the extremely hot summer of 2022, 5011 PWS of the Netatmo brand recorded temperatures, greatly extending the existing coverage of official temperature sensors (see Methods; Supp. Figure 1). PWS are therefore increasingly being used in urban climatological studies and are sought to provide useful complementary data sets for urban climate services¹¹⁻¹⁷”

Presence or absence of: Met Office MIDAS automatic weather station (AWS), Netatmo personal weather stations (PWS), or both

Supp. Figure 1: LSOAs where official weather stations from the UK MetOffice MIDAS network (AWS) are active are in yellow, where *Netatmo* personal weather stations (PWS) are present

are in orange, where both are active are in red (Both) and where none is present are shown in gray (Absence).

In addition, we have plotted the population covered per IMD decile where at least one MIDAS weather station is present and we also found that there is an unequal distribution of official weather sensors with areas most deprived having fewer official weather stations present in their LSOA. This has been added to the supplements as an additional figure:

MIDAS automatic weather station coverage per deciles of Index of Multiple Deprivation

Supp. Figure 2: Percentages of people in England and Wales covered by at least one official automatic weather station (AWS) per LSOA, per Index of Multiple Deprivation (IMD) decile. Most deprived populations are less covered by AWS. The total number of people in each IMD decile is given at the top of the figure and shows fluctuations due to some variability in LSOA population size (~1500 inhabitants per LSOA; see Methods).

We have referred to this figure in the caption of Figure 1:

“**Figure 1:** Percentages of people in England and Wales covered by at least one personal weather station (PWS) per LSOA, per Index of Multiple Deprivation (IMD) decile. Least deprived populations are better covered by PWS. The total number of people in each IMD decile is given at the bottom of the figure and shows fluctuation due to some variability in LSOA population size (~1500 inhabitants per LSOA; see Methods). The same figure is given in Supp. Figure 2 for official automatic weather stations from the MIDAS network handled by

the United Kingdom Met Office and also shows that most deprived populations are less covered by highly accurate weather stations.”

We also added lines 67 to 70:

“Interestingly, we also find that official automatic weather stations from the MIDAS network are less present, if not absent, from most deprived LSOAs (see Supp. Figure 2). At the time of writing, 13.32 % of the population of England and Wales is covered by at least one PWS while only 0.5 % is from a MIDAS weather station. This highlights the extension of weather sensing networks in a variety of environments offered by PWS and supports the need to describe them.”

3. The study is limited to the area of Wales and England. It is not clear why not whole to U.K. was chosen? It seems now a small subset of counties in the home country of the researchers. How this result is generalizable to a wider area remains unclear now. I fully understand it is hard to obtain all socio-economic data across the globe, but at least for Europe this information should be available through Eurostat.

As the reviewer suggests, the study was limited to England and Wales as these two countries offer a dataset at a high level of granularity: the Lower layer Super Output Area (LSOA). For this reason, we are careful not to draw conclusions for places outside of England and Wales. The reason why we did not extend our study to the whole United Kingdom is explained in our discussion and highlighted as a limitation (lines 182 to 185):

“We acknowledge that our study suffers from several limitations and encourage further research to address them. First, its conclusions are specific to England and Wales to benefit from a standardized Index of Multiple Deprivation between the two countries – something that is more complex at the supra-national level (e.g., Europe or global) and even at national level of the countries of the United Kingdom – and should thus be carefully considered for other locations.”

Concerning the comparison to other countries, we agree with the reviewer that this is a great idea and something to aim for. We’d of course like to extend our research to additional countries, but the Indices of Multiple Deprivation and other socio-economic and demographic data that we used in our research is only available for England and Wales. Constructing a variable similar to IMD with the same granularity for inter-country comparison isn’t a trivial job, more particularly since variables that are used to build IMD are generally not available at a similar level to the LSOAs.

In fact, current EUROSTAT statistics related to GDP or population density are only available at NUTS2 or NUTS3, with detailed economic data only available at NUTS2 level. While having the advantage of comprising a unified system across countries, the NUTS2 and NUTS3 geographic areas are quite large and are highly inconsistent in the size of covered areas and population. For example, at NUTS2 level Greater London is divided into five areas, whereas Berlin is a single area, and Paris is just a part of the larger Ile-de-France. If we were to use these levels of administrative unit, our study – which focuses on the presence/absence of PWS to quantify

how much of the population in England and Wales is covered under certain conditions (e.g., levels of deprivation, vegetation density, proportion of ethnic minorities) – would be irrelevant. Indeed, by aggregating our data to these levels we find that all administrative units are covered by at least one PWS.

Supp. Figure 4: Presence of at least one personal weather station (PWS) in Nomenclature of Territorial Units for Statistics (NUTS) administrative units of second (NUTS2) and third (NUTS3) levels.

We do therefore believe that working at larger scale is something to be achieved in future studies and that it will come with its sets of challenges on data aggregation and analysis. We now specifically call for this in the discussion. We have added the above figure in the supplements to support our argument which follows the first limitation of our study in lines 185 to 189:

“Challenges related to the level of aggregation of the socio-economic, demographic and environmental data will have to be addressed to study the factors influencing the distribution of PWS at larger scales. For example, our presence/absence framework would be irrelevant at Nomenclature of Territorial Units for Statistics (NUTS) administrative units as all would be covered by at least one PWS in England and Wales (see Supp. Figure 4)”

4. An additional stratification that could advance the study, is to study whether the same pattern is seen when a) extra instruments are taken into account (i.e. Netatmo the wind and rain sensor are extra costs, so it could be

interesting to know which income threshold or so is needed to buy a wind sensor or rain sensor above the thermometer set), b) instruments from more expensive brands like Davis Vantage Pro (1200 euro instead of 150 euro for a NetAtmo thermometer set).

We agree with the reviewer that it would be interesting to see how annual income plays a role in the acquisition of personal weather stations and how it is correlated to the acquisition of more accurate and expensive devices. We highlighted in our discussion that future studies to gather the information at the individual level (lines 168 to 170):

“[F]uture research should try to understand how different demographic, socio-economic, cultural and environmental covariates may explain the distribution of PWS coverage. Focal population surveys and large-scale machine learning could shed further light on the preliminary findings of our study.”

This would also facilitate research to categorise consumers willing to go for more expensive sensors like the one suggested by the reviewer. We therefore added to the previous sentence (lines 170 to 173):

“For instance, consumer surveys should try to gather information at the individual level to relate how consumers’ demographic and socio-economic characteristics are similar or dissimilar to those of the area in which they live (e.g., distance to the average annual income in the LSOA).”

We haven’t managed however to find any portal to gather the data coming from Davis Vantage Pro sensors. To the best of our knowledge, the only portal that offers publicly accessible weather data nowadays is the Netatmo app developer API. This was written in the limitations of our study (lines 189 to 191):

“[B]ecause of data restriction policies our study only gathered information from Netatmo PWS and did not investigate other crowd-sourced weather data providers like the Weather Observation Website (WOW)⁴⁹ or Weather Underground (Wunderground) which could cover areas deprived of Netatmo PWS⁵⁰.”

5. The study does implicitly imply that the owner of the PWS is also the one who bought the PWS, but PWSes are very often offered as a gift for Christmas, anniversaries and so on. So whether a PWS is present in a garden may depend more about the wealth of the ones who offer the PWS than the ones that operate the PWS in the end. How was dealt with that, and how does this affect your conclusions?

As explained in the methods section, we simply describe where the PWS are located and are working with area-level variables aggregated at the Lower layer Super Output Area (LSOA) and not with individual-level data from the consumers or buyers. We therefore cannot perform this type of study. As explained above we do call for more individual-level data in the discussion.

Reviewer 3

General comments

The main objective of this study is to uncover socio-economic and environmental inequalities in the distribution of crowdsourced personal weather stations. The key findings include a lower presence of crowdsourced personal weather stations (PWS) in deprived environments which can be associated with high build-up, low vegetation cover, high proportions of ethnic minorities and mostly younger people. The study uses straightforward yet methodologically sound empirical techniques to obtain these rather logical but important results.

We thank the reviewer for their valuable comments. Our detailed answers are provided below in black and cited text is provided between quotation marks. We have uploaded the codes and the public data used in our study in a dedicated GitHub repository: https://github.com/oscarbrouse/NatComms_PWS_2024

Major comments

1. How do the authors account for their inability to disentangle the impact of environmental and socio-economic factors (such as built-up density, building height, age, and ethnic minority representation) on PWS coverage independently from the influence of deprivation? What strategies might be employed to disentangle the role of deprivation in explaining PWS coverage?

Several strategies could be adopted to disentangle which variable is the most predictive of PWS coverage, including linear regression models, random forests, gradient boosting or other forms of machine learning. The motivation for such a strategy would be either (1) to predict variations in PWS coverage or (2) to identify the causes of variations in PWS coverage. Regarding motivation (1), there would be no value in predicting as the data we use are complete for the domain of interest and not transferable to another domain. Regarding motivation (2), such an analysis would imply causal hypotheses that are not necessarily logical: we do not believe that variations in built-up density (for example) directly cause variations in PWS coverage; although incomes could potentially be a direct factor. In this work we do not try to explain what causes the distribution of PWS; we are simply observing where PWS are and what type of underlying environments they sense, i.e. what environments are well represented. The last paragraph of our introduction pointed to the main objective of our research. We slightly adapted the last line to put the emphasis on this objective (lines 43 to 50):

“We empirically investigated the current coverage of certain PWS that are widely used by consumers and researchers and that are capable of measuring temperature across England and Wales. [...] Our study provides a comprehensive description of the underlying spatial inequalities in crowdsourced climate data acquisition and accessibility in English and Welsh urban environments.”

In addition, we emphasise the point of concern from the reviewer in the Methods under the “Study’s rationale and framework” section (lines 274 to 280):

“Most importantly, we chose not to perform any explanatory variable analysis because we are not seeking to causally explain the drivers of coverage in PWS, but rather to describe the current state and the implications for sensing of the environment. Moreover, explanation of the drivers of PWS acquisition would be much better addressed by individual data rather than area-level data, for example through surveys or interviews. Machine-learning based studies attempting to explain the distribution of PWS between areas may be obscured by the spatial correlation between PWS coverage and urbanization, which would therefore require extensive studies and analyses that cannot solely be addressed in this paper.”

In other words, our study is the first description of an existing issue that may need to be addressed to cope with the inequalities that we describe in the manuscript. We leave this for future research as multiple methodologies to better understand the drivers of these inequalities and the potential ways forward to mitigate them will have to be developed. Our study is at the frontier of knowledge in crowdsourcing of urban climate data.

2. Why did the authors decide to include the Index of Multiple Deprivation (IMD), Local Climate Zones (LCZs) and a variety socio-economic and environmental indicators, even though they exhibit inherent correlations?

Again, our study does not try to explain what drives the distribution but simply characterise the underlying environments, so the correlation of these variables is not a problem. To describe these environments, we chose a set of easily understood discrete and continuous variables that are relevant for studying urban heat exposure.

3. It is known that coarse vegetation indexes are not ideal to study temperature-environment relationships (Bartesaghi-Koc et al., 2020; Rakoto et al., 2021), since different functional green types may have contradicting cooling effects. In this context, it's important to consider whether the EVI takes into account various functional green types. Could you please clarify how the EVI incorporates information about vegetation density, type, and fraction?

Our study does not try to study the temperature-environment relationship. We here only describe the underlying environment that new personally owned temperature sensors observe. We believe that the Enhanced Vegetation Index (EVI) is an easily understood metric and that it suffices for simply describing whether an LSOA is more or less vegetated. It is a widely known and used vegetation index for studying the level of greenness in vegetated areas. It also has advantages compared to the Normalized Difference Vegetation Index as it reduces the influence of the canopy background signal and the atmosphere, and is more sensitive to the leaf area index and – which is valuable for discretising densely vegetated and treed areas (Son et al. 2014).

We have added lines 244 to 247:

“We chose to use the EVI over the Normalized Difference Vegetation Index as it reduces the influence of the atmosphere and the canopy background signal on the measured index while having an increased sensitivity to the leaf area index – hence facilitating the discretisation of different vegetation types in high biomass locations (like forests)^{62,63}.”

Son, N. T., Chen, C. F., Chen, C. R., Minh, V. Q., & Trung, N. H. (2014). A comparative analysis of multitemporal MODIS EVI and NDVI data for large-scale rice yield estimation. *Agricultural and forest meteorology*, 197, 52-64.

Liao, Z., He, B., & Quan, X. (2015). Modified enhanced vegetation index for reducing topographic effects. *Journal of Applied Remote Sensing*, 9(1), 096068-096068.

4. There appears to be missing information regarding the LCZ map used in the study. Was the study area extracted directly from a broader European LCZ map? Can a Europe-wide LCZ map provide sufficient precision for examining the specific LCZs within England and Wales? Additionally, it would be valuable to know more about the spatial resolution, accuracy, and other relevant characteristics of the LCZ map employed in the research.

We thank the reviewer for noticing that some valuable information was missing about the European LCZ map. We did already explain in the methods that we used this map for our study. It has already been widely applied across Europe for urban climate studies and is sought to be applicable across the whole continent. We have added some information about the resolution and the accuracies to showcase the latter in lines 253 to 256:

“The LCZ were mapped at a resolution of 100 m with an overall accuracy above 80 % for the whole of Europe making it a valid land use land cover map for studying urban climate related issues in the region; the overall accuracy, the Kappa score and the true positive rates for evaluating how the urban areas’ extension were consistent with other products like ESA CCI are also above 90 % for the UK⁶⁴.”

5. Given that PWSs have demonstrated the capability to capture fine-scale temperature variations, encompassing both inter- and intra-LCZ variability (Fenner et al., 2017; Verdonck et al., 2017), it might be considered a limitation to classify PWSs solely into a major LCZ category based on the predominant LCZ within the corresponding Lower Layer Super Output Area (LSOA). Such an approach may result in the loss of valuable information. It would be beneficial to include details regarding the number of distinct LCZ types within a single LSOA and their respective proportions, providing a more nuanced perspective on the urban microclimate in the study area.

We fully agree with the reviewer that classifying the LSOAs based on the modal LCZ can lead to a reduction in information; something intrinsic to any classification work. Nonetheless, we argue that for the purpose of this study – describing the underlying environments of PWS’ locations in England and Wales – such a simplification can be pursued. For instance, Stewart and Oke (2012) describe Local Climate Zones as “*regions of uniform surface cover, structure, material, and human activity that span hundreds of meters to several kilometers in horizontal scale.*” (pp. 1884). We therefore believe that the aforementioned uniformity is well captured by the modal LCZ when it comes to describing the environment in which ~1500 inhabitants live (LSOA) as the latter will capture the most influential LCZ on the local climate. This question of uniformity under the LCZ scheme is dependent on the goal of the research. Urban climate modellers often use similar simplifications (e.g., modal or averages of LCZ parameters) to model the local climate at kilometer scales. If we were to predict temperatures at the LSOA level, then proportions of LCZ would be valuable.

We have added our rationale for using only the modal LCZ under a Supplementary Information section that is referred to in the Methods (lines 267 to 268):

“[U]sed the same package to get the modal LCZ in the LSOA (see Supp. Information 2 for more information on the potential loss of information induced by the use of modal LCZ)”

We have added the supplementary information section 2 entitled “Modal LCZ and loss of information”:

“Supplementary Information 2: Modal LCZ and loss of information

Characterising each Lower layer Super Output Area (LSOA) by a single Local Climate Zone necessarily comes with a simplification of the complexity of the environment that composes each LSOA; something natural to any classification exercise. In this study, we chose to classify each LSOA in the form of Local Climate Zone by choosing the modal LCZ as the determining entity of the LSOA's *urban climatic* background. The modal LCZ is the LCZ that composes most of the LSOA area (Supp. Figure 5). This could therefore come with an over simplification of the LSOA environment in cases where, for example, LSOAs are composed of two major LCZs that are only separated by a few percentage points in their proportion of the covered area (e.g., 48 % vs 52 %). Below, we show how choosing the modal LCZ as a class for the entire LSOA is expected to have a low impact on the conclusions presented in this study.

In England and Wales, as higher population densities are found around cities, urban LSOAs tend to be smaller than their natural counterparts. This minimises the chances of having multiple disparate LCZs within urban LSOAs as can be seen in Supp. Figure 5 (e.g., panels **a** to **f** and **h**). As population density lowers in rural areas (Supp. Figure 5 panels **g** and **f** to **o**) LSOAs cover larger areas and are therefore more prone to a plurality of LCZs that compose their territory; despite the hegemony of *LCZ D: Low Plants* (Supp. Figure 5I) in rural England and Wales.

Supp. Figure 5: Proportions of each LCZ composing the land-use land-cover of England and Wales in each Lower layer Super Output Area (LSOA). LCZs are extracted from the European LCZ map from Demuzere et al. (2019). For the LCZ codification, please refer to the main manuscript or Supp. Figure 6, below.

By plotting the cumulative density function of proportions of other LCZ in each modal LCZ at the LSOA level (Supp. Figure 6), we find that other LCZs do not compose more than a 1/3 of the LSOA area in more than 80 % of the cases (except for LCZ 6 in LSOAs classified as LCZ 3, LCZ 8 and LCZ B (Supp Figure 6b, e and h), remembering that LCZ 6 ($n=22529$) composes most of the LSOAs in England and Wales with LCZ D ($n=8027$). This shows that in a vast majority the modal LCZ is unanimously the one that composes the larger part of each LSOA and can therefore be expected to have the greatest influence on the local climate and characterise it. Withal, other prominent LCZs in each LSOA are of similar types than the modal LCZ in which they are embedded. For example, urban *Compact Low-Rises* (LCZ 3) are generally found in greater proportions in LSOAs classified as *Compact Mid-Rises* (LCZ 2; Supp. Figure 6a). This means that independent of the presence of other LCZs in each LSOA, a generic typology of local climate can be expected, such as: compact urban, open urban, sparsely built and natural (forested or afforested).

Supp. Figure 6: Cumulative distributions of proportions of other LCZs composing Lower layer Super Output Areas (LSOAs) classified as LCZ_x using a modal classification. Distributions are from 50 bin histograms spanning proportions from 0 % to 50 %; above 50 %, the LCZ would necessarily become modal. The horizontal light dashed line indicates that half of LCZ_x are lower or greater than a proportion p . The horizontal bold dashed line indicates the same but with 80 % of the proportions being lower than p . The vertical dashed line shows a proportion p equal to 1/3 of the LSOA area.

Our results are in line with previous studies from Bechtel et al. (2017) which showed that there is a greater chance of having similar LCZ clustered together; this supports the development of the weighted accuracy for evaluating LCZ maps' accuracy. By considering "how wrong" our modal classification is, to cite Bechtel et al. (2020), we argue that our modal classification certainly misses part of the LSOAs' climatic environment complexity. At the same time; the expected climate from the modal LCZ is not thought to be entirely dissimilar to one that would consider the variety of LCZs that compose each LSOA. For the purpose of our study, which is simply to characterise the typology of local climates that are currently being sensed at the national scale, we believe that the modal LCZ is therefore sufficient. Other studies that would try to predict air temperature at the LSOA level or to investigate the causal explanations of the personal weather station density are encouraged to use more discrete variables like the proportion of LCZ.

Reference

Bechtel, B., Demuzere, M., Sismanidis, P., Fenner, D., Brousse, O., Beck, C., ... & Verdonck, M. L. (2017). Quality of crowdsourced data on urban morphology—the human influence experiment (HUMINEX). *Urban Science*, 1(2), 15.

Bechtel, B., Demuzere, M., & Stewart, I. D. (2020). A weighted accuracy measure for land cover mapping: comment on Johnson et al. local climate zone (LCZ) map accuracy assessments should account for land cover physical characteristics that affect the local thermal environment. *Remote Sens.* 2019, 11, 2420. *Remote Sensing*, 12(11), 1769.

Demuzere, M., Bechtel, B., Middel, A., & Mills, G. (2019). Mapping Europe into local climate zones. *PloS one*, 14(4), e0214474.“

Stewart, I. D., & Oke, T. R. (2012). Local climate zones for urban temperature studies. *Bulletin of the American Meteorological Society*, 93(12), 1879-1900.

6. Building upon the preceding comment, as the entire dataset is aggregated at the LSOA level, it is reasonable to assume that readers would find it valuable to have insight into the characteristics and key metrics of these LSOAs. Providing an overview of the LSOA layout and sharing general metrics for these areas can enhance the reader's understanding of the study context.

We believe this is exactly what we do in the study. We indeed provide details about the LSOA characteristics and maps of each metrics used in the study at the LSOA level. We also describe what an LSOA is in the text.

7. The organization of the Results and Methods sections can be improved for better readability. To achieve this, incorporating subheadings to break down the content into smaller, more manageable sections would be beneficial. Additionally, aligning the flow and structure of both sections could further enhance the overall readability of the manuscript.

We agree with the reviewer's suggestion. We have added some subheadings to the Results, the Discussion and the Methods sections which can be found in the updated manuscript. We would like to keep the structure as is concerning the Methods section to be more in line with Nature Communications' publication style: coming after the Discussion.

Minor comments

8. Line 12-13: This sentence should be supported with appropriate citations: "Particular attention has been given to urban heat because temperature is a reliably measured weather variable, known to have negative impacts on health, buildings, energy, or biodiversity."

We agree with the reviewer. We have added these references:

Murage, P., Hajat, S., Macintyre, H. L., Leonardi, G. S., Ratwatte, P., Wehling, H., ... & Kovats, S. (2023). Indicators to support local public health to reduce the impacts of heat on health. *Environment International*, 108391.

Tomlinson, C. J., Chapman, L., Thornes, J. E., & Baker, C. J. (2011). Including the urban heat island in spatial heat health risk assessment strategies: a case study for Birmingham, UK. *International journal of health geographics*, 10(1), 1-14.

Dessai, S., Fowler, H. J., Hall, J. W., & Mitchell, D. M. (2022). UK Climate Risk Assessment and Management. *Climate Risk Management*, 37.

Kaiser, A., Merckx, T., & Van Dyck, H. (2016). The Urban Heat Island and its spatial scale dependent impact on survival and development in butterflies of different thermal sensitivity. *Ecology and Evolution*, 6(12), 4129-4140.

Hinkel, K. M., Nelson, F. E., Klene, A. E., & Bell, J. H. (2003). The urban heat island in winter at Barrow, Alaska. *International Journal of Climatology: A Journal of the Royal Meteorological Society*, 23(15), 1889-1905.

9. Figure 2: From the caption of Figure 2 it is unclear whether the total surface and total population of the different LCZ classes in figure 2a and figure 2b relate to the original LCZs derived from WUDAPT or whether they present the LSOAs with modal LCZ.

The population is derived from the LSOAs and we simply sum up the total amount of population living in the LSOA by the modal LCZ. WUDAPT never offered any population estimates. We believe that this was clear in our previous version.

10. Figure 2: Reference to panel b is missing in the figure caption: "Personal weather stations (PWS) per square kilometre (km²; a) and percentage of total population covered by a PWS (km²; b) in each Local Climate Zone (LCZ)."

Checked and dealt with.

11. Line 105-106: "Also, higher albedo are observed by ~0.02 (2 % more reflective capacity), EVI by ~0.06 (denser vegetation), and proportion of people aged 65 or above by ~10 %". This sentence gives the impression that the most deprived LSOAs are associated with higher albedo, denser vegetation and older people. That seems quite counterintuitive and the opposite is shown in Figure 4:

We thank the reviewer for noticing this. We have rewritten the paragraph following previous comments made by the other reviewers. It now reads (lines 110 to 119):

"These demographic and environmental characteristics of sensors deserts also describe lower IMD deciles (Figure 4, Supp. Table 1). For example, most deprived LSOAs have higher building heights by ~4 m than least deprived LSOAs on average (Supp. Table 1), building fractions by ~10 %, and proportion of ethnic minorities by ~15 %. Also, lower albedo is observed by ~0.02 (2 % less reflective capacity), EVI by ~0.06 (less densely vegetated), and proportion of people aged 65 or above by ~10 %.

No deserts are found at similar levels of deprivation

Importantly, we find that within the population living in each IMD decile no clear distinction between the PWS and no-PWS can be made out of the environmental and demographic indicators. For example, the highest proportion of people living in the highest

IMD decile (10th) live in the 3rd decile of building height (low buildings) independent of whether PWS are present or absent in their LSOA (Figure 4, Supp. Table 1).”

12. Line 112: It is not indicated where the results of the PSS are shown. A reference to Supplementary Table 1 could be included.

Checked and dealt with.

13. Line 127: Period punctuation between sentences is missing.

Checked and dealt with.

14. Line 132: Period punctuation between sentences is missing.

We do not see what the reviewer is referring to.

15. Line 140-142: “Furthermore, the sensor desert may limit quantification of the cooling induced by vegetated evapotranspiration and related benefits to heat mitigation (Lungman, et al, 2023).” The statement is true but the cited reference does not consider PWS in quantifying the cooling potential of vegetation.

We agree with the reviewer. This now reads (lines 153 to 155):

“Furthermore, the sensor desert may limit quantification of the cooling induced by vegetated evapotranspiration, the latter being shown to have positive impacts on heat mitigation and related heat-mortality⁴⁵.”

16. Line 142-145. This sentence is quite long. Consider breaking it down into smaller parts to improve the readability: “Moreover, as people aged above 65 tend to be more susceptible to heat-related health hazards and because ethnic minorities are often more exposed to asphalted built-up areas and their related surface urban heat islands – which could drive disparities in heat exposure – it would be necessary to adequately quantify how vulnerabilities induced by deprivation may increase the vulnerability of different communities, stratified by age groups.”

Checked and dealt with. This now reads (lines 153 to 159):

“Furthermore, the sensor desert may limit quantification of the cooling induced by vegetated evapotranspiration, the latter being shown to have positive impacts on heat mitigation and related heat-mortality⁴⁵. Lastly, ethnic minorities are often more exposed to asphalted built-up areas and related surface urban heat islands^{46,47}. This could drive disparities in heat exposure, especially considering that ethnic minorities tend to comprise younger populations and that people aged above 65 tend to be more susceptible to heat-related health hazards²⁹. It would therefore be necessary to adequately quantify how vulnerabilities induced by deprivation may increase the risk to different communities, stratified by age groups.”

17. Line 156-157: The closing parenthesis seems to be missing: “For instance, it would be necessary to disentangle how deprivation explains PWS coverage because other covariates (e.g., built-up density, population density, or building height may also have an influence on PWS coverage that we could not separate from deprivation in this study.”

Checked and dealt with.

18. Line 193: This sentence appears to be incomplete: “We use a moving window of 0.2° by 0.2° over England and.”

Checked and dealt with. We added “Wales”.

19. Line 261: This reference appears to be incomplete: “Oke, T. R., Mills, G., Christen, A. & Voogt, J. A. Urban climates (Cambridge University Press, 2017)”

The reference is appropriate and refers to a book. The formatting comes from the OverLeaf template provided by Nature Communications. We will check this with the editors once the paper is in production as some valuable information is missing in this formatting. We have added the DOI to the reference:

```
“@book{oke2017urban,  
title={Urban climates},  
author={Oke, Timothy R and Mills, Gerald and Christen, Andreas and Voogt, James A},  
year={2017},  
isbn={9781139016476},  
url={https://doi.org/10.1017/9781139016476},  
publisher={Cambridge University Press}  
}”
```

20. Line 318: This reference appears to be incorrect/incomplete: “for National Statistics, O. Office for national statistics: 2011 census geography products for england and wales.”.

Checked and dealt with. We also added to the *bibtex* the date of access:

```
“@online{ONS2011,  
title={Office for National Statistics: 2011 Census geography products for England and Wales},  
author={{Office for National Statistics}},  
url={https://webarchive.nationalarchives.gov.uk/ukgwa/  
20160105225829/http://www.ons.gov.uk/ons/guide-method/geography/products/census/index.html},  
year={2011},  
addendum = {(accessed: 17.03.2023)}  
}”
```

21. Line 319-320: This reference appears to be incorrect/incomplete: “Schaaf, C. & Wang, Z. Mcd43a3 modis/terra+ aqua brdf/albedo daily l3 global–500m v006. nasa eosdis land processes daac (2015).”

We agree with the reviewer. We have dealt with the formatting of this reference and others in the *bibtex* to keep capital letters adequately displayed.

```
“@article{schaaf2015mcd43a3,
```

```
title={{MCD43A3 MODIS/Terra+ Aqua BRDF/Albedo Daily L3 Global--500m V006. NASA EOSDIS Land Processes DAAC}},
author={Schaaf, C and Wang, Z},
journal={US Geological Survey: Reston, VA, USA},
year={2015},
url={https://doi.org/10.5067/MODIS/MCD43A3.006}
}"
```

References

- Bartesaghi-Koc, C., Osmond, P., & Peters, A. (2020). Quantifying the seasonal cooling capacity of 'green infrastructure types' (GITs): An approach to assess and mitigate surface urban heat island in Sydney, Australia. *Landscape and Urban Planning*, 203(February), 103893. <https://doi.org/10.1016/j.landurbplan.2020.103893>
- Fenner, D., Meier, F., Bechtel, B., Otto, M., & Scherer, D. (2017). Intra and inter 'local climate zone' variability of air temperature as observed by crowdsourced citizen weather stations in Berlin, Germany. *Meteorologische Zeitschrift*, 26(5), 525–547. <https://doi.org/10.1127/metz/2017/0861>
- Rakoto, P. Y., Deilami, K., Hurley, J., Amati, M., & Sun, Q. (Chayn). (2021). Revisiting the cooling effects of urban greening: Planning implications of vegetation types and spatial configuration. *Urban Forestry and Urban Greening*, 64(January), 127266. <https://doi.org/10.1016/j.ufug.2021.127266>
- Verdonck, M. L., Okujeni, A., van der Linden, S., Demuzere, M., De Wulf, R., & Van Coillie, F. (2017). Influence of neighbourhood information on 'Local Climate Zone' mapping in heterogeneous cities. *International Journal of Applied Earth Observation and Geoinformation*, 62(September 2016), 102–113. <https://doi.org/10.1016/j.jag.2017.05.017>

REVIEWERS' COMMENTS

Reviewer #1 (Remarks to the Author):

In this paper, the authors show that crowdsourced weather data have the potential to improve climate adaptation and that it is therefore important to understand the biases in personal weather stations' distribution.

The methodology is simple, yet sound and appropriate.

The authors have successfully taken my previous review into account, and I have no further comment.

Reviewer #2 (Remarks to the Author):

Evaluation: In my view the authors have sufficiently addressed the comments of the reviewers. The research has been thoroughly executed, and has an important and new message for science and society. The research has some limitations in terms of scale and data availability, but these have been well discussed, and recommendations for future steps have been well formulated. Hence these limitations should not be a showstopper in publishing the manuscript.

Recommendation: accept for publication

Reviewer #3 (Remarks to the Author):

The authors have thoroughly addressed all the comments raised during the 1st revision. I have no additional feedback to provide at this time.